# Multicenter Real-World Data of Subsequent Chemotherapy after Progression to PARP Inhibitors in a Maintenance Relapse Setting

**DOI:** 10.3390/cancers14184414

**Published:** 2022-09-11

**Authors:** Margarita Romeo, Marta Gil-Martín, Lydia Gaba, Iris Teruel, Álvaro Taus, Claudia Fina, Maria Masvidal, Paola Murata, Julen Fernández-Plana, Alejandro Martínez, Cristina Pérez, Yolanda García, Valerie Rodriguez, Sara Cros, Marta Parera, Montserrat Zanui, Silvia Catot, Beatriz Pardo, Andrea Plaja, Anna Esteve, Maria Pilar Barretina-Ginesta

**Affiliations:** 1Medical Oncology Department, Institut Català d’Oncologia Badalona, Badalona Applied Research Group in Oncology (BARGO), Institut d’Investigació Germans Trias i Pujol (IGTP), Carretera del Canyet s/n, 08916 Badalona, Spain; 2Medical Oncology Department, Institut Català d’Oncologia L’Hospitalet, Hospital Duran i Reynals, IDIBELL, Gran Via 199-203, 08909 L’Hospitalet de LLobregat, Spain; 3Medical Oncology Department, Hospital Clínic de Barcelona, Translational Genomics and Targeted Therapies in Solid Tumors, IDIBAPS, Carrer Villarroel 170, 08036 Barcelona, Spain; 4Medical Oncology Department, Hospital del Mar-CIBERONC, Cancer Research Program, IMIM (Hospital del Mar Medical Research Institute), Passeig Marítim 25-29, 08003 Barcelona, Spain; 5Medical Oncology Department, Institut Català d’Oncologia Girona, Girona Biomedical Research Institute IdIBGi, Av França s/n., 17007 Girona, Spain; 6Medical Oncology Department, Hospital Universitari de Reus, Avda Josep Laporte 2, 43204 Reus, Spain; 7Medical Oncology Department, Hospital Universitari Arnau de Vilanova, Av. Alcalde Rovira Roure, 80, 25198 Lleida, Spain; 8Medical Oncology Department, Hospital Universitari Mutua de Terrassa, Plaça Dr Robert 5, 08221 Terrassa, Spain; 9Medical Oncology Department, Hospital Quirón Dexeux, Sabino de Arana 5-19, 08028 Barcelona, Spain; 10Medical Oncology Department, Xarxa Sanitària i Social Santa Tecla, C/Rambla Vella, 14-16 4, 43003 Tarragona, Spain; 11Medical Oncology Department, Hospital Universitari Parc Taulí, Institut d’Investigació i Innovació Parc Taulí (I3PT), Universitat Autònoma de Barcelona, Carrer Parc Tauli 1, 08208 Sabadell, Spain; 12Medical Oncology Department, Hospital Verge de la Cinta, Carrer de les Esplanetes 44, 43500 Tortosa, Spain; 13Medical Oncology Department, Hospital General de Granollers, Avda Francesc Ribas s/n, 08402 Granollers, Spain; 14Medical Oncology Department, Hospital Universitari de Vic, Carrer de Francesc Pla el Vigatà, 1, 08500 Vic, Spain; 15Medical Oncology Department, Hospital de Mataró, Carretera de la Cirera 230, 08304 Mataró, Spain; 16Medical Oncology Department, ALTHAIA, Xarxa Assistencial Universitària de Manresa, Dr. Joan Soler 1-3, 08243 Manresa, Spain; 17Oncology Data Analytics Program (ODAP), Medical Oncology Department, Institut Català d’Oncologia Badalona, Badalona Applied Research Group in Oncology (BARGO), Institut d’Investigació Germans Trias i Pujol (IGTP), Carretera del Canyet s/n, 08916 Badalona, Spain

**Keywords:** ovarian cancer, PARP inhibitors, platinum sensitivity, platinum rechallenge, mechanisms of resistance

## Abstract

**Simple Summary:**

Since the irruption of PARPi in the therapeutic armamentarium for ovarian cancer, concerns regarding post-progression treatment outcomes have emerged, owing to known crossed-resistance mechanisms between PARPi and platinum. In this multicentric retrospective series of ovarian cancer patients, we evaluated chemotherapy results upon progression to maintenance with PARPi in the relapsed setting. We further selected the population of platinum-sensitive patients (according to the classical definition) retreated with platinum (*n* = 74). In this platinum-sensitive population, overall response rate and survival outcomes of platinum rechallenge after PARPi were similar to historical series of the prePARPi era. However, within this group, analysis according to BRCA status showed that BRCA mutant patients (*n* = 35) presented higher rates of progression and worse survival outcomes under subsequent platinum than BRCA wild type patients (*n* = 39), with statistically significant differences. This is the largest real-world data series of ovarian cancer patients treated with platinum rechallenge in the post-PARPi scenario.

**Abstract:**

Background: Despite impressive progression-free survival (PFS) results from PARP inhibitors (PARPi) in ovarian cancer, concerns about their effect on post-progression treatment outcomes have recently arisen, particularly when administered in the relapsed setting. Overlapping mechanisms of resistance between PARPi and platinum have been described, and optimal therapies upon progression to PARPi are unknown. We communicate real-world data (RWD) on outcomes of subsequent chemotherapy upon progression to PARPi used as maintenance in ovarian cancer relapses, particularly focusing on platinum rechallenge, according to BRCA status. Methods: Data from high-grade serous or endometrioid ovarian cancer patients who received subsequent chemotherapy after progression to maintenance PARPi in the relapsed setting, in 16 Catalan hospitals between August 2016 and April 2021, and who were followed-up until July 2021, were included. Endpoints were overall response rate (ORR), and PFS and overall survival (OS) measured from the subsequent chemotherapy starting date. Results: 111 patients were included [46 (41.4%) presented pathological BRCA1/2 mutations, 8 (7.5%) in other homologous recombination-related genes]. Sixty-four patients (57.7%) had received two prior chemotherapy lines, including the one immediately prior to PARPi. PARPi were niraparib (*n* = 60, 54.1%), olaparib (*n* = 49, 44.1%), and rucaparib (*n* = 2, 1.8%). A total of 81 patients remained platinum-sensitive (PS population) after progression to PARPi (when progression-free interval [PFI] was >6 months after the last cycle of prior platinum) [median PFI 12.0 months (interquartile range, IQR, 8.8–17.1)]. Of those, 74 were treated with subsequent platinum regimens, with the following results: ORR of 41.9%, median PFS (mPFS) of 6.6 months (95% CI 6–9.2), and median OS (mOS) of 20.6 months (95% CI 13.6–28.9). Analysis of these 74 patients according to BRCA status showed that PFIs for BRCA mutant and non BRCA-mutant patients were 13.6 [IQR11.2–22.2] and 10.3 [IQR 7.4–14.9] months, respectively (*p* = 0.010); ORR were 40.0% versus 43.6%, respectively; Rates of progression (as best response) to subsequent platinum were 45.7% versus 17.9%, respectively (*p* = 0.004); mPFS and mOS were 3.5 (95% CI 2.5–8.6) versus 7.5 months (95% CI 6.5–10.1, *p* = 0.03), and 16.4 (95% CI 9.3–27.5) versus 24.2 months (95% CI 17.2–NR, *p* = 0.036), respectively. Conclusion: This is the largest series of real-world data on ovarian cancer patients retreated with platinum in the post-PARPi scenario, separately analyzing BRCA mutant and non-mutant patients, to our knowledge. In our platinum-sensitive population, rechallenge with platinum after progression upon PARPi in the 3rd or later lines for ovarian cancer relapses shows relevant ORR and similar PFS outcomes to historical series of the prePARPi era. However, BRCA mutant patients presented significantly higher rates of progression under subsequent platinum and worse survival outcomes associated with subsequent platinum than non-BRCA-mutant patients.

## 1. Introduction

Ovarian cancer is the leading cause of gynecologic cancer mortality in the US and the EU, despite the decrease of its mortality rate by 6.6% between 2017 and 2022, partly because of the use of oral contraceptives and the reduced use of hormone replacement therapy [1]. Its traditional median overall survival (OS) of around 50% at 5 years has been achieved with the forefront combination of debulking cytoreductive surgeries and platinum-based doublets, as well as chemotherapy rechallenge for subsequent relapses or progressions. The use of platinum in subsequent lines largely depends on the expected sensitivity to this drug, and classically, platinum-sensitive relapses have been defined as relapses occurring >6 months after the last cycle of prior platinum (progression-free interval [PFI] >6 months), at least up to the third relapse [2,3]. Improving OS of ovarian cancer patients is a priority for oncologists dedicated to gynecologic cancers.

Fortunately, the better molecular characterization of epithelial OC with the identification of BRCA1 and 2 mutations, mutations in other DNA damage repair genes, and the homologous recombination (HR) deficient status, paved the way for the introduction of PARP inhibitors (PARPi) [4,5]. Also, platinum sensitivity emerged early on as a key factor for a greater PARPi benefit [6]. In 2014, the European Medicaments Agency (EMA) granted olaparib with the first PARPi indication in oncology, which was as maintenance therapy after response to platinum-based chemotherapy for BRCA mutant platinum-sensitive relapsed ovarian cancer patients, based on the impressive progression-free survival (PFS) results from Study 19 in this molecular subgroup compared to placebo [7]. This trial showed 11.2 months versus 4.3 months (HR 0.18, *p* < 0.0001) in favor of olaparib among BRCA mutant patients, in a retrospective subgroup analysis according to BRCA status. These results were confirmed in the phase 3 trial SOLO2, including only the BRCA mutant population, which showed a median PFS (mPFS) of 19.1 months in the olaparib arm versus 5.5 months with placebo arm [8]. Later, PFS results from large phase III/IV trials (NOVA/ENGOT-OV16 [9], ARIEL3 [10], OPINION [11], ORZORA [12]) supported further indications of niraparib, rucaparib, and olaparib as maintenance therapy after platinum used in the platinum-sensitive relapsed setting regardless of molecular status. Importantly, overall, these trials showed that the magnitude of benefit from PARPi after response to platinum-based chemotherapy differs among BRCA-mutant, HR-deficient or HR-proficient populations [13]. In the NOVA trial, the only one in which BRCA wild type (BRCAwt) patients constituted a population sized to explore statistically significant differences with placebo, the median PFS for niraparib was 9.3 months versus 3.8 for placebo in this molecular subgroup (HR 0.45, *p=* 0.001) [14]. Neither olaparib, niraparib nor rucaparib in the relapsed setting have shown statistically significant increases in OS, despite the first-mentioned showing a 12.9-month extension in OS with respect to the placebo in SOLO2 trial with exclusively BRCA-mutant population [15].

Olaparib was also the first PARPi that showed positive results in the adjuvant/first-line setting, concretely with 2 years of maintenance treatment for the BRCA mutant population (SOLO1 trial) [16]. Its impressive results enabled nearly 50% of the patients in the olaparib arm to be free of relapse at 5 years of follow-up [17]. Recently, niraparib and the combination of bevacizumab + olaparib have broadened indications in the first line for other molecular subgroups, again with different magnitudes of benefit [18,19]. In these trials, PARPi set the median PFS in the first line ranging 14–22 months for the overall population, which means that more than 50% of OC patients will progress while on PARPi in the first line. Burning questions are whether OS will eventually increase in the overall population and whether the benefit is greater in the first-line or in the recurrence setting for the BRCAwt patients.

Therefore, many patients will eventually progress upon PARPi and mechanisms of resistance to these drugs are being intensively investigated. The best characterized are BRCA secondary reversion mutations, which can restore BRCA functionality (and thus HR pathway) in patients harboring germline original BRCA mutations [20]. They are known to confer crossed resistance to both PARPi [21] and platinum [22]. ARIEL4 provided a prospective report from a randomized trial demonstrating that their presence predicted primary resistance to PARPi, showing a median PFS of 2.9 months in the rucaparib arm (*n* = 13) vs. 5.5 months in the chemotherapy arm (*n* = 10) [23]. These mutations are present in approximately 25% of the cases, arising under therapeutic pressure with platinum and/or PARPi. Despite the fact that they are the only validated mechanisms of resistance to PARPi in the clinical arena, their identification has not been implemented in the clinics yet [24]. Other crossed-resistance mechanisms have been described in the preclinical and clinical arenas, such as epigenetic reversions of hypermethylated BRCA1 promoter or CCNE1 amplifications [20,25,26], but their incidence is unknown. Therefore, despite a significant prolongation of PFS in the first- and second-line settings shown in all PARPi pivotal trials, early doubts aroused on the effectiveness of post-PARPi rechallenge with platinum. Current guidelines abrogate for a flexible algorithm when electing a chemotherapy schema at relapse, taking into consideration response to prior platinum, treatment-free interval for platinum, number of prior lines and persistent toxicity (among others) [5], but PFI is still a major parameter that defines expected platinum sensitivity in real life and in clinical trials. Deeper knowledge about the mechanisms of resistance to PARPi will allow for deciding on the optimal therapy after progression to PARPi [20].

The aim of the current study is to assess clinical outcomes of chemotherapy treatments used upon progression to PARPi (olaparib/rucaparib/niraparib) as maintenance after platinum therapy for ovarian cancer relapses, regardless of BRCA mutations, in a real-world data (RWD) setting. Specifically, we aim to focus on the results of subsequent platinum-based regimens in the population remaining platinum-sensitive (PS) after progression to PARPi, and according to BRCA status (mutant versus non-mutant).

## 2. Materials and Methods

### 2.1. Study Design

This is a retrospective multicentric study including high-grade serous or endometrioid ovarian cancer patients that have progressed to PARPi used as maintenance therapy after platinum in the relapsed setting and who have received subsequent chemotherapy as per clinical practice in 16 Catalan hospitals between August 2016 and April 2021. To be included in the study, at least one radiological evaluation of the subsequent chemotherapy line had to be reported, according to investigator criteria. Patients were followed up until July 2021. Upon the request of local ethical committees, informed consent was obtained from patients.

The retrieved information from each patient was age at diagnosis, histologic subtype (high-grade serous or high-grade endometrioid), BRCA mutational status (pathogenic mutant *versus* wild type or variants of unknown significance), presence of mutations in other homologous recombination-related (HRR) genes if available, treatment-free interval between last cycle of prior platinum to initiation of platinum immediately pre-PARPi (prior TFI), number of previous lines (2 vs. >2), details regarding platinum-based chemotherapy prior to PARPi (schemas, cycles and months on treatment), best response to this chemotherapy (complete response vs. partial response or, if any, stabilizations), progression-free interval since the last cycle of platinum pre-PARPi (PFI), details regarding PARPi therapy (type, months on treatment, and use of PARPi in the first line), progression-free interval since the beginning of PARPi (PFI-PARPi), details regarding subsequent chemotherapy (schemas, cycles and months on treatment), best response to subsequent chemotherapy (complete, partial, stabilization or progression), date of progression, and survival information. Database was locked on 15 July 2021. Follow-up was calculated from the date of subsequent chemotherapy initiation to last contact or date of death.

For our purpose of evaluating the results of chemotherapy after PARPi, and specifically platinum rechallenge, we defined two populations according to the PFI after the last cycle of platinum immediately pre-PARPi [2,3]. Therefore, we identified patients with a platinum-sensitive (PS) relapse after progression to PARPi (our main interest) as those with PFI > 6 months, and patients with a platinum-resistant (PR) relapse after progression to PARPi as those with PFI < 6 months. In the PS-population treated with platinum-based chemotherapy after progression to PARPi, we further explored 2 subgroups according to BRCA status (Figure 1).

### 2.2. Outcomes

ORR was described as the percentage of patients with complete or partial response to the subsequent chemotherapy line after PARPi.

PFS was defined as the time from the date of initiation of the subsequent chemotherapy line after PARPi until the date of progression or death from any cause.

OS was defined as the time from the date of initiation of the subsequent chemotherapy line after PARPi until the date of death from any cause.

### 2.3. Statistical Analysis

Continuous data were summarized using median and interquartile ranges (IQRs) and categorical data with counts and percentages; moreover, both were compared between groups of patients using the Mann–Whitney U test and the chi-square test, respectively. OS and PFS curves were estimated using the Kaplan-Meier statistics and were compared between strata using the log-rank test. BRCA mutational status, number of previous lines to PARPi, best response to the platinum line immediately previous to PARPi, and PFI were included in multivariate Cox regression models to estimate the effect on the OS and PFS hazard ratios (HRs) and 95% confidence intervals (CIs) were reported. Patients who entered OREO study (maintenance with PARPi rechallenge vs. placebo) [27] after subsequent chemotherapy were excluded from all survival analyses. All analyses were performed using R software v4.1.1 (Vienna, Austria).

## 3. Results

### 3.1. Patients’ Characteristics and General Description of the Populations of Study

During the study period, 111 patients were included in a centralized database. Subsequent chemotherapy lines after PARPi were initiated between 25 August 2016 and 1 April 2021, which we would consider our study period. Overall, 81 patients progressed beyond 6 months after the last cycle of platinum immediately previous to PARPi (PS-population), and 30 progressed within the first 6 months (PR-population) (see flowchart in Figure 1). Median follow-up of the whole sample was 11.5 months (IQR 5.1–17.8).

The main baseline characteristics and details of prior therapies in the PS- and PR- populations were very similar, as shown in Table 1, except for the distribution of BRCA mutant patients. Overall, the median age at diagnosis was 59.8 years old (53.9–69.4), and the most frequent histology was high-grade serous (96.4%). Forty-six patients (41.4%) harbored pathological BRCA1/2 mutations (40 germline, six somatic, thereafter “BRCA mutant patients”). Among the 59 non BRCA-mutant patients, 7 presented mutations in other HRR genes, and 6 had BRCA1/2 variants of unknown significance. Remarkably, one patient presented a pathogenic mutation in BRCA2 and a probable pathogenic mutation in CHEK2, both germline. Specifically, BRCA mutant patients accounted for 44.4% in the PS subpopulation (36 out of 81) and 33.3% in the PR subpopulation (10 out of 30).

Regarding prior therapies, 64 patients (57.7%) had received 2 prior chemotherapy lines (including the one immediately previous to PARPi) and 47 patients (42.3%) had received more than 2 prior lines. The most frequent chemotherapy regimen immediately before PARPi was platinum-doublets (*n* = 100, 90.1%). Responses to this line of therapy were as follows: 21 complete responses (18.9%), 81 partial responses (73.0%), and 9 stabilizations (8.1%). Niraparib and olaparib were the most-used PARPi (*n* = 60, 54.1%, and *n* = 49, 44.1%, respectively), and only 2 patients had received rucaparib (1.8%). Only 1 patient had received PARPi during the first-line setting (inside clinical trial) before receiving it in the recurrent setting again (plus 3 patients unknown allocated in a placebo or PARPi arm).

In the PS population, median PFI was 12.0 months (IQR 8.8–17.1), and median PFI-PARPi was 9.9 months (IQR 6.2–14.6). In the PR population, median PFI was 4.6 months (IQR 3.7–5.2 months), and median PFI-PARPi was 3.0 months (IQR 2.4–3.9 months).

### 3.2. Outcomes in the PS Population

Of the 81 PS patients, 74 received subsequent platinum-based regimens immediately after PARPi and 7 received subsequent non-platinum-based regimens, as shown in Flowchart 1.

Among those who received subsequent platinum, platinum-doublets without bevacizumab were the most-used regimens (*n* = 52, 70.3%) (see Figure 1). Median duration of platinum-based regimens was 6 cycles (IQR 3–6), and 3.9 months (IQR 1.8–5.3). Importantly, 5 entered the maintenance trial OREO (maintenance with PARPi rechallenge vs. placebo) at the end of this chemotherapy line, and therefore they were excluded from PFS and OS analyses but included in the ORR analysis. Results of subsequent platinum after PARPi in this population were as follows: ORR of 41.9%, median PFS (mPFS) of 6.6 months (95% CI 6–9.2), and median OS (mOS) of 20.6 months (95% CI 13.6–28.9), as shown in Table 2. Results did not change when we excluded the 5 patients with stabilization as best response to platinum-based chemotherapy prior to PARPi.

When describing BRCA mutant and non BRCA-mutant populations, we only found differences in the PFI, in the PFI-PARPi and the type of PARPi. Median PFI was 13.6 [IQR 11.2–22.2] months in the BRCA mutant group versus 10.3 [IQR 7.4–14.9] in the non-mutant group (*p* = 0.010). Median PFI-PARPi was 12.2 [IQR 9.1; 20.0] months in the BRCA mutant group versus 8.3 [IQR 5.7; 12.7] in the non-mutant group (*p* = 0.006). As expected, considering historical and regional approvals, olaparib was the predominant drug among BRCA mutant patients (94.3%) versus niraparib among the non-mutant patients (79.5%, *p* = 0.001). For further information related to BRCA status subgroups, see Table 3. As shown in Table 2, analysis according to BRCA status showed that the ORR of BRCA mutant and non-mutant patients were 40.0% and 43.6%, respectively. Interestingly, progression rates as best response to subsequent platinum were 45.7% and 17.9%, respectively (*p* = 0.004). Regarding survival outcomes, mPFS and mOS were 3.5 (95% CI 2.5–8.6) and 16.4 months (95% CI 9.3–27.5), respectively, in the BRCA mutant group versus 7.5 (95% CI 6.5–10.1, *p* = 0.03) and 24.2 months (95% CI 17.2–NR, *p* = 0.036), respectively, in the non BRCA-mutant group (see Figure 2). Results did not change when we further excluded 5 patients with stabilization as best response to platinum-based chemotherapy prior to PARPi (apart from the 5 patients already excluded for having participated in OREO study).

Outcomes were not significantly different statistically according to PFI subgroups, those with a partially PS relapse (PFI 6–12 months) (*n* = 33) versus those with fully PS disease (PFI> 12 months) (*n* = 41): ORR were 33.4% versus 48.8% (*p* = 0.271), mPFS were 6.1 [95% CI 4.9–9.9] versus 7.2 [95% CI 6.5–10.8] months (*p* = 0.593), and mOS were 17.2 [95% CI 13-NA] versus 20.6 [95% CI 13.2-NA] months (*p* = 0.933), respectively. These results and a full description of baseline characteristics, details of prior lines and details of subsequent chemotherapy after PARPi of these 2 subgroups are shown in Appendix A.

We explored whether BRCA status, number of previous lines (2 or more), best response to the platinum line immediately preceding PARPi (CR versus PR/stabilization), or PFI subgroups were potentially related to higher ORR and longer survival outcomes. None of the studied prognostic factors showed a particular effect on ORR, but BRCA status was the only significant factor for PFS and OS in the univariate and multivariate analysis, showing that non-BRCA-mutant patients harbored a better prognosis with respect to those in the BRCA mutant group. In the multivariate Cox regression model for PFS, the BRCA mutant group had an estimated HR of 2.60 (95% CI 1.39–4.86, *p* = 0.003). Similarly, the multivariate model for OS showed an estimated HR of 2.89 (95% CI 1.31–6.38, *p* = 0.009).

Additionally, among these 74 PS patients treated with subsequent platinum, we identified 20 patients who met the OREO trial inclusion criteria related to the required duration of first PARPi at relapse in this trial (12 months for BRCA mutant patients or 6 months for non-mutant patients), and the duration of the subsequent platinum (at least 4 cycles). As stated above, only 5 out of 20 finally entered OREO trial, while 15 did not. These 20 patients accounted for 24.7% of the PS population. Distribution according to known mutations was 9 BRCA mutant (45%), 3 harbored mutations in other HRR genes (15%), and 8 did not harbor any of these mutations (40%). Excluding the 5 patients who effectively entered OREO, mPFS to subsequent platinum-based chemo in this subgroup was 10.8 (95% CI 8.2-NR) months, and mOS was 20.6 (95% CI 13.6-NR) months.

The subsequent non-platinum-based regimens were trabectedin plus pegylated liposomal doxorubicin (*n* = 3) and non-platinum monotherapy (*n* = 4) (see Figure 1). Median duration of platinum-free regiments was 4 cycles (IQR 4–4.5), and 3.2 months (IQR 2.6–4.1). Specific arguments alleged by physicians to justify the use of a platinum-free regimen in the PS-population were prior toxicity or allergy to platinum, or progressive shortening of PFI in prior lines. This subgroup was enriched in non-BRCA-mutant patients (85.7%), none of them presented complete response to the platinum immediately prior to PARPi, and all of them were in the category PFI 6–12 months. See full description of this subgroup of patients in Appendix A. Treatment with platinum-free regimens in this population showed 14.3% of ORR, mPFS of 3.8 months (95% CI 3-NR), and mOS of 10.4 months (95% CI 4.6-NR).

### 3.3. Outcomes in the PR Relapsed Population

Of the 30 PR patients, 13 received subsequent platinum-based regimens immediately after PARPi, and 17 received subsequent platinum-free regimens, as shown in Flowchart 1. The main argument explaining why physicians opted for platinum-based regimens in these cases was that treatment-free interval was >6 months (median 6.4 [95% CI 5.3–7.3]), in spite of PFI < 6 months. Most of the platinum-based regimens were doublets (76.9%, *n* = 10), while all platinum-free regimens consisted of a single-agent chemotherapy +/− bevacizumab. The median duration of platinum-based regimens was 4 cycles (IQR 4–6), and 3.5 months (IQR 2.3–3.7). The median duration of platinum-free regimens was 6 cycles (IQR 4–8), and 4.8 months (IQR 3.0–7.0). Ten patients (33.3%) were in the BRCA mutant group, six in the group of subsequent platinum-based regimens and four in the group of platinum-free regimens.

Results of subsequent platinum after PARPi among the PR-population were as follows: ORR of 46.2%, mPFS of 4.7 months (95% CI 3.4–NR), and mOS of 10.1 months (95% CI 6.3–NR). Results did not significantly differ between BRCA-mutant and non-BRCA-mutant subgroups. In comparison, platinum-free regimens in this population showed 47.1% of ORR (*p* = 0.9), mPFS of 6.8 months (95% CI 6.8–NR, *p* 0.144), and mOS of 14.4 months (95% CI 10.7–NR, *p* = 0.496). These results are summarized in Appendix A.

## 4. Discussion

In this retrospective series of 111 ovarian cancer patients treated with chemotherapy upon progression to PARPi, used as maintenance after platinum-based regimens for relapses, we have specifically focused on results from platinum rechallenge in those patients with PFI > 6 months. In other words, we have addressed the issue of rechallenge with platinum after progression during PARPi therapy in platinum-sensitive ovarian cancer patients according to the classic definition used in clinical trials. In this subgroup of 74 patients, rechallenge with platinum, which represented the 3rd line of therapy for most of our patients, offered an ORR of 41.9%, mPFS of 6.6 months, and mOS of 20.6 months. Interestingly, we observed a particularly poor mPFS of subsequent platinum among BRCA mutant patients (3.5 months), in contrast to the non-BRCA-mutant population (7.5 months, *p* = 0.03), as well as a poorer mOS. We also found a significantly higher progression rate to platinum rechallenge among the BRCA mutant subgroup (45%) in comparison to the non-BRCA-mutant subgroup (17.6%, *p* = 0.004), despite similar ORR between them. Overall, this is the largest published series of real-world data on ovarian cancer patients retreated with platinum in the post-PARPi scenario, separately analyzing BRCA-mutant and non-mutant patients, to our knowledge.

The interest of this topic aroused when some overlapping mechanisms of resistance between PARPi and platinum salts were described in the in vitro and in vivo arenas. Despite the impressive results of PARPi as maintenance after platinum response for relapse, clinical concerns have grown over time regarding a potential decreasing of benefits of platinum rechallenge after progression to PARPi.

First attempts to study post-progression outcomes in the randomized pivotal trials of PARPi in ovarian cancer were the secondary endpoints PFS to subsequent therapy (PFS2) and time to second subsequent therapy (TSST), both censoring patients who have not yet progressed to PARPi. Results of these outcomes have favored olaparib and rucaparib over placebo, showing statistically significant increases both in the BRCA-mutant population [15] and in the all-comers population [10].

However, the first clinical evidence of a potential detrimental effect of PARPi on subsequent platinum-based chemotherapy came from a post-hoc analysis of SOLO2 trial communicated in ESMO congress 2020 by Frenel et al., including only BRCA1/2-mutant patients who had progressed to olaparib/placebo [28]. This work, very recently published, showed a longer time to second progression (TSP, measured from progression to olaparib/placebo to progression to subsequent chemotherapy line) among patients who had received placebo in comparison to those treated with olaparib, particularly when subsequent chemotherapy was platinum-based (14.3 months versus 7 months, *n* = 42 and 54, respectively) [29]. Additionally, little data is available from a post-hoc analysis of NOVA trial showing no differences of PFS2-PFS1 (defined as the time between progression to niraparib/placebo to the subsequent progression or death) between the placebo and niraparib arms [30], which could be consistent with a potential balanced effect between BRCA and non BRCA-mutant population. However, in ARIEL3, the same post-hoc analysis found no differences between the placebo and rucaparib arms in the BRCA mutant, HR deficient or the overall population [10].

In 2012, Hanker et al. published an article that set historical references about the impact of the second to the sixth line of therapy on survival of relapsed ovarian cancer after primary taxane/platinum-based therapy. This work was based on data of 1620 patients from three large, randomized phase III trials investigating primary therapy that were conducted by two large European collaborative groups from 1995 to 2002. They found that a maximum of three lines of subsequent treatment seemed to be beneficial in terms of OS for patients with recurrent ovarian cancer, and that platinum sensitivity remained an independent prognostic factor in the first and further relapses. Specifically, they found that mPFS of patients treated with chemotherapy at the second, third, and fourth relapses were 7.2, 6.5, and 4.7 months, respectively, and that mOS were 14.2, 10.6, and 7.7 months, respectively [3]. Focusing on our general outcomes from platinum rechallenge for platinum-sensitive relapses after PARPi, it seems that our mPFS is comparable to data from the 3rd lines of chemotherapy in this historical pre-PARPi series (outcomes for second relapses). On the other hand, our mOS result is clearly better than that historically reported, probably suggesting improvements in all therapeutic areas including increasing available experimental therapies for our patients in the last decade. One of the strong points of our work relies on a strict definition of platinum-sensitivity, which favors our comparison to data from these historical clinical trials from the pre-PARPi era. However, there are differences in the type of chemotherapy and patients’ selection between our series and this historical work. Notwithstanding, this comparison allows us to quickly evaluate our results from a bird’s eye view.

Very little real-world data of the post-PARPi scenario has been published so far. In an Italian series reporting data from maintenance olaparib in 234 relapsed BRCA-mutant patients (>50% of them in the 3rd or later lines), response to subsequent therapies were evaluable in 66 patients. Among the 18 patients with a platinum-free interval of >12 months (14 of whom retreated with post-PARPi platinum), ORR was 22%. Patients with a platinum-free interval of 6–12 months (*n* = 27, approximately 36% of them treated with subsequent platinum) had an ORR of 11.1%. ORR of patients with a platinum-free interval of <6 months (*n* = 21, all of them treated with non-platinum therapies) was 9.5% [31]. According to their reported response distribution, 51% of BRCA mutant patients with a platinum-free interval of > 6 months presented progression as best response to subsequent post-PARPi chemotherapy, which is in line with our findings. In another recent Italian publication of a retrospective series of 103 patients (46% BRCA-mutant) treated with chemotherapy after progression to PARPi for maintenance at relapse (only 34 with subsequent platinum), patients obtained response rates of 13%, 26%, and 42% in the subgroups with a platinum-free interval of <6 months (*n* = 23), 6–12 months (*n* = 42), and >12 months (*n* = 31), respectively. mOS for these subgroups were 8.9, 17.5, and 24.1 months, respectively. Our mOS regardless of BRCA status are consistent with these results [32].

Another approach to evaluate post-progression platinum using real-world data is to compare the PARPi treated cohort with historical cohorts or independent series without PARPi. The largest study using this methodology is a multicenter observational retrospective study comparing BRCA mutant patients treated with olaparib as maintenance in the second line versus those who had not received olaparib (historical control group) in the same setting. Considering those who received platinum-based chemotherapy as the third line, median PFS2-PFS1 in the olaparib group (*n* = 33) was 8.9 months, while it was 14.8 months in the control group (*n* = 62) (*p* = 0.023). Data from patients that had not progressed to PARPi were censored at 1 day. Results were similar when considering non-platinum subsequent chemotherapies, suggesting that resistance to olaparib may contribute to overall chemotherapy resistance in BRCA mutant patients. The variable PFS1 below or over 12 months did not show a significant result in the multivariable analysis, in line with our results [33]. In another series of 92 patients retreated with platinum in the 3rd line of treatment, 35 had received prior PARPi and 57 had not. Progression rates to platinum in the 3rd line were 40% and 9%, respectively. However, data of BRCA status are not reported [34]. Overall, these data suggest that PARPi may have a negative impact on the effectiveness of subsequent platinum, at least in the BRCA-mutant population.

Therefore, in the platinum-sensitive population of our study, we found that mPFS of platinum rechallenge in the 3rd or later lines is similar to historical pre-PARPi data from large clinical trials, while mOS is consistent with more modern series that analyzed the benefit of post-PARPi chemotherapy lines. Our findings according to BRCA status suggest that PARPi may have a negative impact on subsequent platinum results in the BRCA population (consistently with prior reports), in comparison to non BRCA-mutant population. Of note, median PFI-PARPi was longer among the BRCA mutant subgroup (as expected), while its mPFS associated with platinum rechallenge was very poor (much worse than TSP reported by Frenel et al., which was calculated similarly to our PFS). Due to this poor result in our BRCA-mutant population, we could assume these patients have exhibited crossed resistance to both PARPi and platinum compounds, probably in part owing to the emergence of BRCA reversion mutations while on prior PARPi. Conceptually, mechanisms of resistance to PARPi occur due to restoration of HR, restoration of the replication fork stability, PARP alterations, or they are related to multidrug resistance mechanisms. While BRCA reversion mutations are specific to the BRCA-mutant population, other less-known mechanisms may be specific to the non-BRCA-mutant population, but their incidence is mostly unknown [20].

Notwithstanding, in the subgroup of patients that would have potentially accomplished main inclusion criteria for OREO trial (that is 25% of the PS population treated with platinum rechallenge, approximately half of them BRCA mutant in our series), we found longer mPFS to subsequent platinum after PARPi. OREO is a positive proof-of-concept trial that evaluated the rechallenge with PARPi as a second maintenance after response to a subsequent platinum after first progression to PARPi. Only patients with a duration of PARPi in the relapse setting for more than 6 months or 12 months in BRCA wild type or BRCA mutant groups, respectively, could enter the trial. Despite selecting this population, the benefits of PARPi rechallenge resulted clinically marginal, but translational research is expected to provide biomarkers to identify a small subset of patients that obtained a prolonged benefit from PARPi rechallenge, up to 37 months. Again, results of PARPi rechallenge were slightly better among BRCA wild type patients [27].

Several therapeutic strategies are being investigated in the post-PARPi scenario, such as a combination of PARPi with WEE1 inhibitors (EFFORT trial) [35], antiangiogenics (EVOLVE trial) [26], or ATR inhibitors (CAPRI trial) [36], among others. EVOLVE has been the first of them published, showing a mPFS of 5.4, 7.6, and <2 months for patients with baseline mutations in HRR genes, those without, and those with reversion mutations in these genes or upregulation of ABCB1, respectively [26]. These phase 2 trials conducted in a heavily pretreated heterogeneous population (PS and PR relapsed patients, BRCA mutant and BRCAwt) are accompanied by ambitious translational programs (often with mandatory biopsy at inclusion) that will enable the oncological community to optimize post-PARPi therapies.

We must acknowledge some limitations of our study, the first of which is its retrospective nature. In this sense, when evaluating outcomes of platinum rechallenge in the PS and PR populations, we must bear in mind that our sample is enriched patients who received PARPi in late lines according to emerging regional indications during the study period. Additionally, BRCA-mutant patients accounted for 41.4% of the whole sample, a much higher proportion than the presence of BRCA1/2 mutations at diagnosis of ovarian cancer patients (~25%); this fact may suggest a selection bias to receive subsequent chemotherapy (and even PARPi), probably derived from the natural evolution of both diseases (cancers harboring BRCA1/2 mutations and cancers not harboring these mutations). Second, due to our design, the exclusion of patients who have not progressed to PARPi may have selected a population with particularly worse outcomes, especially among BRCA-mutant patients. In fact, our median PFI-PARPi in BRCA-mutant patients was 12.2 months, consistent with mPFS of olaparib in Study 19 [7], but much shorter than mPFS of PARPi in the pivotal phase 3 trials for this molecular subgroup (around 20 months) [8,37]. Considering this potential bias and the fact that platinum rechallenge for PS relapses is a standard in gynecologic oncology based on randomized trials [38], we cannot conclude that BRCA mutant patients should not be retreated with platinum after progression to PARPi in spite of our poor results in this molecular population; instead, our work puts a word of caution in this topic and supports the hypothesis of molecularly characterizing these patients upon progression to PARPi, as well as the idea of implementing the identification of BRCA reversion mutations in the clinical scenario of early acquired PARPi resistance. On the other hand, our median PFI-PARPi in non-BRCA-mutant patients was 8.3 months, consistent with mPFS data from Study 19 and pivotal phase 3 trials (around 8 months in this molecular subgroup) [7,14,37]. Therefore, our results of platinum rechallenge after maintenance PARPi in the non-BRCA-mutant population could be generalized to this molecular population in the relapsed setting and maybe to a significant part of this population progressing while on PARPi in the first line setting.

## 5. Conclusions

In conclusion, mechanisms of resistance to PARPi are being intensively investigated to optimize subsequent therapies. These mechanisms will probably vary between the BRCA mutant and wild-type populations and may also differ depending on the duration of prior therapies. Despite the fact that rechallenge with platinum may still be a potentially useful therapy after PARPi, our data show that caution must be taken when considering its use in the BRCA-mutant population progressing earlier than expected. Partial overlap of mechanisms of resistance between platinum and PARPi may have a major impact on this population. Characterizing these mechanisms is a relevant unmet need in our daily practice as this could allow for the identification of molecular subpopulations for which new experimental therapies may be crucial.

## Figures and Tables

**Figure 1 cancers-14-04414-f001:**
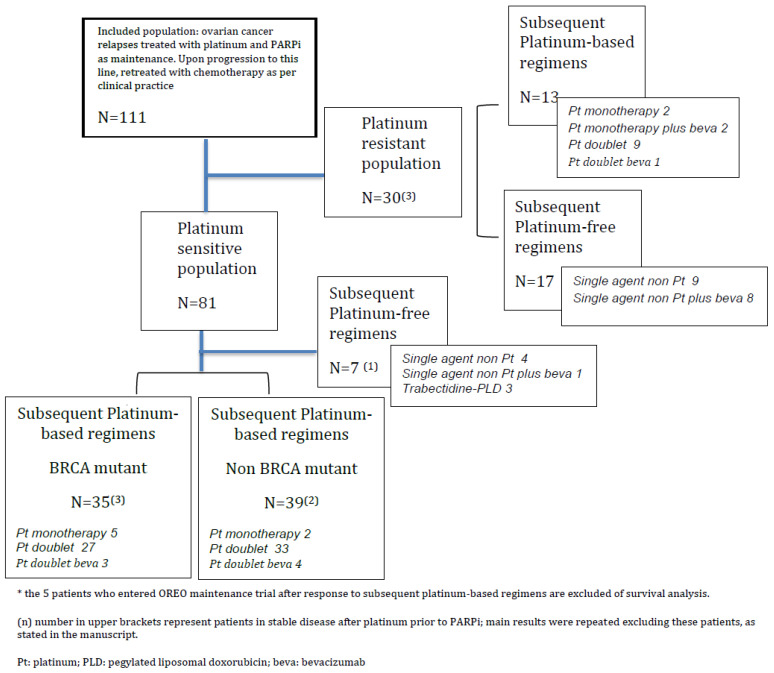
Flow chart of the study and schemas of subsequent chemotherapy after PARPi.

**Figure 2 cancers-14-04414-f002:**
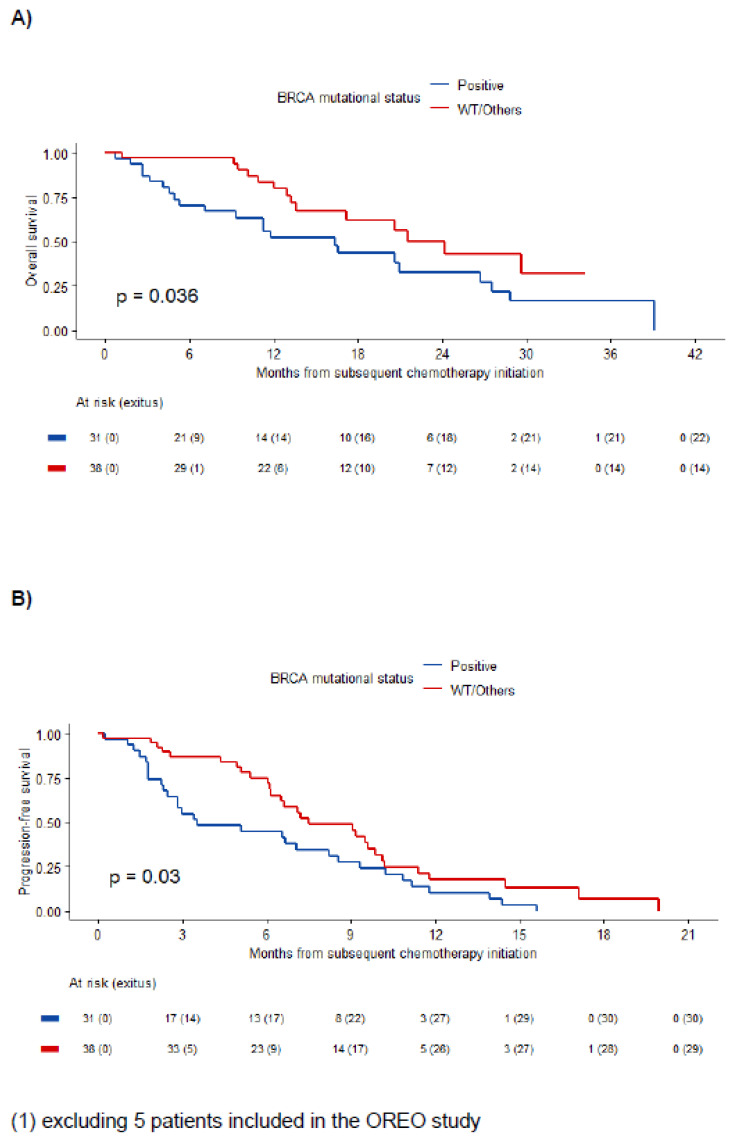
Kaplan-Meier estimates of OS (**A**) and PFS (**B**) in PS patients receiving subsequent platinum by BRCA mutation (1).

**Table 1 cancers-14-04414-t001:** Baseline characteristics and previous treatments in PS and PR patients.

	Overall	PS	PR	*p*
	*n* = 111	*n* = 81	*n* = 30	
Age at diagnosis (years), Median [IQR]	59.8 [53.9; 69.4]	60.0 [55.7; 69.2]	58.1 [52.5; 70.3]	0.725
Histology, *n* (%):				0.573
HGSOC	107 (96.4%)	77 (95.1%)	30 (100.0%)	
HGEOC	4 (3.6%)	4 (4.9%)	0 (0.0%)	
BRCA mutational status, *n* (%):				0.131
Positive	46 (41.4%)	36 (44.4%)	10 (33.3%)	
WT	59 (53.2%)	39 (48.1%)	20 (66.7%)	
Variants of unknown significance (VUS)	6 (5.4%)	6 (7.4%)	0 (0.0%)	
Other HRR genes mut, *n* (%):				-
Positive	9 (9.6%)	8 (11.3%)	1 (4.3%)	
Pt-based chemo immediately previous to PARPi, *n* (%):				1.000
Pt monotherapy	11 (9.9%)	8 (9.9%)	3 (10.0%)	
Pt doublet	100 (90.1%)	73 (90.1%)	27 (90.0%)	
Months from last cycle of prior Pt to initiation of Pt immediately pre-PARPi (prior TFI), *n* (%):				0.019
<6 m	3 (2.7%)	0 (0.0%)	3 * (10.0%)	
6–12 m	46 (41.4%)	33 (40.7%)	13 (43.3%)	
>12 m	58 (52.3%)	46 (56.8%)	12 (40.0%)	
Non-available	4 (3.6%)	2 (2.5%)	2 (6.7%)	
Number of previous lines to PARPi (including the immediately previous), Median [IQR]	2.0 [2.0; 3.0]	2.0 [2.0; 3.0]	2.0 [2.0; 3.8]	0.267
Number of previous lines to PARPi (including the immediately previous), *n* (%):				0.730
=2	64 (57.7%)	48 (59.3%)	16 (53.3%)	
>2	47 (42.3%)	33 (40.7%)	14 (46.7%)	
Number of cycles of Pt-based chemo immediately previous to PARPi, Median [IQR]	6.0 [5.0; 6.0]	6.0 [5.0; 6.0]	6.0 [5.0; 6.0]	0.072
Best response to Pt-based chemo immediately previous to PARPi, *n* (%):				0.319
CR	21 (18.9%)	18 (22.2%)	3 (10.0%)	
PR	81 (73.0%)	57 (70.4%)	24 (80.0%)	
SD	9 (8.1%)	6 (7.4%)	3 (10.0%)	
Months with Pt-based CT treatment, Median [IQR]	4.3 [3.5; 5.1]	4.6 [3.6; 5.5]	4.1 [3.4; 4.7]	0.055
PARPi, *n* (%):				0.008
olaparib	49 (44.1%)	42 (51.9%)	7 (23.3%)	
niraparib	60 (54.1%)	37 (45.7%)	23 (76.7%)	
rucaparib	2 (1.8%)	2 (2.5%)	0 (0.0%)	
Months with PARPi, Median [IQR]	7.4 [4.3; 12.2]	9.5 [6.5; 14.7]	3.7 [2.8; 4.0]	<0.001
Prior PARPi, *n* (%):				0.388
Yes	1 (1.0%)	1 (1.3%)	0 (0.0%)	
No	99 (96.1%)	74 (97.4%)	25 (92.6%)	
Maybe	3 (2.9%)	1 (1.3%)	2 (7.4%)	
Months from finishing Pt pre-PARPi to Pt pre-PARPi progression (PFI), Median [IQR]	9.3 [5.6; 14.2]	12.0 [8.8; 17.1]	4.6 [3.7; 5.2]	<0.001
Months from PARPi initiation to Pt pre-PARPi progression (PFI-PARPi), Median [IQR]	7.5 [4.0; 12.3]	9.9 [6.2; 14.6]	3.0 [2.4; 3.9]	<0.001

Pt: platinum; PS: platinum-sensitive; PR: platinum-resistant. * in spite of prior TFI < 6 months, patient’s physician considered that these 3 patients would benefit from rechallenge with platinum at relapse and after this new platinum-based line they eventually prescribed PARPi as maintenance.

**Table 2 cancers-14-04414-t002:** ORR, PFS and OS in PS patients receiving subsequent platinum by BRCA mutation.

	Total	BRCA MUTANT	BRCA WT/Others	*p*
	*n* = 74	*n* = 35	*n* = 39	
Best response to CT administered after PARPi progression, *n* (%):		0	0.004	
CR	6 (8.1%)	5 (14.3%)	1 (2.6%)	
PR	25 (33.8%)	9 (25.7%)	16 (41.0%)	
SD	20 (27.0%)	5 (14.3%)	15 (38.5%)	
PROG	23 (31.1%)	16 (45.7%)	7 (17.9%)	
OS *	20.6 (13.6–28.9)	16.4 (9.3–27.5)	24.2 (17.2, NR)	0.036
PFS *	6.6 (6–9.2)	3.5 (2.5–8.6)	7.5 (6.5–10.1)	0.030

* excluding patients included in the OREO study.

**Table 3 cancers-14-04414-t003:** Baseline characteristics and previous treatments in PS patients receiving subsequent platinum by BRCA mutation.

	Mutant	WT/VUS	*p*
	*n* = 35	*n* = 39	
Age at diagnosis (years), Median [IQR]	59.7 [53.5; 65.7]	61.2 [56.5; 70.6]	0.155
Histology, *n* (%):			0.117
HGSOC	35 (100.0%)	35 (89.7%)	
HGEOC	0 (0.0%)	4 (10.3%)	
Pt-based chemo immediately previous to PARPi, *n* (%):			0.245
Pt monotherapy	5 (14.3%)	2 (5.1%)	
Pt doublet	30 (85.7%)	37 (94.9%)	
Months from last cycle of prior Pt to initiation of Pt immediatly pre-PARPi, *n* (%):			0.553
6–12 m	13 (37.1%)	18 (46.2%)	
>12 m	21 (60.0%)	21 (53.8%)	
Non-available	1 (2.9%)	0 (0.0%)	
Number of previous lines to PARPi (including the immediately previous), Median [IQR]	2.0 [2.0; 3.0]	2.0 [2.0; 3.0]	0.493
Number of previous lines to PARPi (including the immediately previous), *n* (%):			0.423
=2	23 (65.7%)	21 (53.8%)	
>2	12 (34.3%)	18 (46.2%)	
Number of cycles of Pt-based chemo immediately previous to PARPi, Median [IQR]	6.0 [6.0; 6.0]	6.0 [5.0; 6.0]	0.477
Best response to Pt-based chemo immediately previous to PARPi, *n* (%):			0.579
CR	10 (28.6%)	8 (20.5%)	
PR	22 (62.9%)	29 (74.4%)	
SD	3 (8.6%)	2 (5.1%)	
Months with Pt-based CT treatment, Median [IQR]	4.7 [4.0; 5.8]	4.2 [3.3; 5.3]	0.055
PARPi, *n* (%):			<0.001
olaparib	33 (94.3%)	7 (17.9%)	
niraparib	2 (5.7%)	31 (79.5%)	
rucaparib	0 (0.0%)	1 (2.6%)	
Months with PARPi, Median [IQR]	11.9 [8.2; 20.7]	8.3 [5.8; 12.7]	0.007
Prior PARPi, *n* (%):			1.000
No	32 (100.0%)	36 (97.3%)	
maybe	0 (0.0%)	1 (2.7%)	
Months from finishing Pt pre-PARPi to Pt pre-PARPi progression, Median (PFI) [IQR]	13.6 [11.2; 22.2]	10.3 [7.4; 14.9]	0.010
Months from PARPi initiation to Pt pre-PARPi progression, Median (PFI-PARPi) [IQR]	12.2 [9.1; 20.0]	8.3 [5.7; 12.7]	0.006

Pt: platinum.

## Data Availability

Data available on request due to restrictions for privacy and ethical requirements. The data presented in this study are available on request from the corresponding author. The data are not publicly available due to ethical requirements.

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
