# Peer review of "Multicenter Real-World Data of Subsequent Chemotherapy after Progression to PARP Inhibitors in a Maintenance Relapse Setting"

_cancers, 2022, doi:10.3390/cancers14184414_

Round 1

Reviewer 1 Report

The authors present Multicenter real-world data of subsequent chemotherapy after progression to PARP inhibitors in a maintenance relapse setting. The paper has meaningful results for prognostic factor of platinum re-challenge after PARPi maintenance in recurrent ovarian cancer. The treatment strategy after recurrence of PARPi maintenance is the one of the unmet need in recurrent ovarian cancer. However, there are some issues that could be clarified or explained more completely. I have some comments and suggestions:

Major points;

1.Authors conclude that BRCA mutant patients presented significantly higher rates of progression under subsequent platinum and worse survival outcomes by multivariate analysis. HRD status is also important for PARPi maintenance, so patients with HRR related gene should be considered for univariate and multivariate analysis.

2. There were no figures and tables in manuscript.

Minor points

Abstract:

Line 92  preiPARP -> prePARPi

Introduction

Line 123;  reference ?

Discussion

Line 381;  reference ?

Author Response

Dear Reviewer 1,

Thank you for having read our work and arising interesting points that could be improved. Please find below our response (in italics) to your comments (in bold). (Note: all mentioned lines in Responses belong to the version with track-changes visible of the resubmitted manuscript).

Major points

1.Authors conclude that BRCA mutant patients presented significantly higher rates of progression under subsequent platinum and worse survival outcomes by multivariate analysis. HRD status is also important for PARPi maintenance, so patients with HRR related gene should be considered for univariate and multivariate analysis.

RESPONSE: We agree with the reviewer that HRD status is also important and thus HRR related gens should be considered. However, we did not have complete information about them. The reason is that, in our specific geographical area, BRCA status is routinely analyzed for therapeutic purposes because it is the major predictive factor of response to PARPi, while other HRR genes are not routinely analyzed because germline panels vary among different hospitals and have changed over time. To clarify this point, we have modified the text in page 4 line 190 (“if available”).

  1. There were no figures and tables in manuscript.

RESPONSE: We are not sure of what may have happened. Now, we have made sure that the revised versions of them are uploaded (main Tables and Figures and Supplementary). Please, don’t hesitate to tell us if they are not available neither. 

Minor points

Abstract:

Line 92; preiPARP -> prePARPi

RESPONSE: Corrected. Thank you for picking up this typo error!

Introduction

Line 123; reference?

RESPONSE: We have added a reference for each of the trials mentioned in this line. Thank you for pointing at this gap, we think now the manuscript is more complete.

Discussion

Line 381;  reference?

RESPONSE: We have corrected this reference in page 13 (reference 29).

Please, don’t hesitate to contact us for further clarifications. Yours sincerely,

Reviewer 2 Report

The authors reported the clinical outcomes of chemotherapy treatment used upon progression to PARPi as maintenance therapy for ovarian cancer relapses in real-world data ( RWD).  Since, post-PARPi is one of the hot topics in this era and not many papers upon PARPi RWD has been published, the study is intriguing.

Major;

There are several major and minor comments that the authors should give answers.

1, In simple summary, the authors wrote that BRCA mutants patients (n=35) presented higher rates of progression and worse survival outcomes. From the main results we can see that this results are from platinum-sensitive group, but the authors should clarify the sentence. It is confusing whether the result is from platinum-sensitive+ platinum-resistant group or in other subgroups. In addition, the authors described that patients with BRCAmut has worse prognosis when subsequent platinum therapy was followed after PARPi in platinum- sensitive group. In addition, did the prognosis of the patients among BRCAwt and BRCAmut differ when subsequent platinum therapy was followed in the platinum-resistant group ?(not the statistical results but only the description of the trend is ok, since on 13 used subsequent platinum therapy in the platinum-resistant group).

2. It is not clear who received the PARPi. It is suggested that the authors write the inclusion criteria of who received PARPi. In addition, it is suggested whether the HRD status were analyzed. The authors wrote “other HRR genes” and it is suggested to clarify those HRR genes. Also, please clarify whether the status of harboring the HRR genes was one of the indications of receiving PARPi and the related reference to include those patients with HRR genes.

3. In Supple Table2, the authors wrote “BRCA stauts-> others (n=4 in total)”. Please clarify who the “others” were.

4. Please describe how many prior chemotherapy lines were pursued before the PARPi. In addition, please describe when each patient became platinum resistant; the patients could have become platinum resistant after 1st line of chemotherapy, or second line of platinum based-chemotherapy after the 1st platinum based chemotherapy was sensitive. The results may differ from the ones became platinum-resistant after 1st platinum based chemotherapy and 2nd platinum based chemotherapy.

5. It seems the detailed information about the patients are necessary, therefore it is suggested that the raw data of patients’ clinico-genetic information be submitted.

6. Page 6 line 274-282, the authors described the figure 2. It seems that the figure 2 presents the survival curve of the patients without the 5 patients who were included in OREO study. However, page 6 line 274-281 seems to describe the ones with the 5 patients who were included in OREO study. It is suggested to clarify the sentences.

7. Page 6, line 304, the authors wrote “we identified 20 patients who could have potentially entered the OREO trial after response to subsequent platinum~~”.It means that the authors do not know who exactly were involved in the OREO trial and among them, who took “PARPi” or “placebo” as a maintenance treatment. The total of patients included in this study were 111 and 20 pts are about 18% of the total number, and it is not small. Therefore, it is suggested to identify the patients’ data related to OREO study and if it were not possible, it may be better to exclude the 20 patients.

8. Page 7. Line 324, the authors wrote that platinum-based regimens in platinum-resistant patients were used because the treatment-free interval was >6 months. Usually, “platinum-sensitive” is considered when the treatment-free interval was >6 months therefore, what the authors wrote in the sentence is not clear. Please, describe the definition of platinum-sensitive and platinum-resistant in this study.

Minor:

1. Page2, line92: preiPARPàpre PARPi
2. Page6, line 277”: p 0.004
à p=0.004

Author Response

Dear Reviewer 2,

Thank you for having read our work and arising interesting points that could be improved. Please find below our response (in italics) to your comments (in bold). (Note: all mentioned lines in Responses belong to the version with track-changes visible of the resubmitted manuscript).

Major:

There are several major and minor comments that the authors should give answers.

  1. In simple summary, the authors wrote that BRCA mutants patients (n=35) presented higher rates of progression and worse survival outcomes. From the main results we can see that this results are from platinum-sensitive group, but the authors should clarify the sentence. It is confusing whether the result is from platinum-sensitive+ platinum-resistant group or in other subgroups. In addition, the authors described that patients with BRCAmut has worse prognosis when subsequent platinum therapy was followed after PARPi in platinum- sensitive group.

RESPONSE: Thank you for raising this point. We have modified the Simple summary to clarify that main results explained in this section belong to the platinum-sensitive group (lines 50-53).

In addition, did the prognosis of the patients among BRCAwt and BRCAmut differ when subsequent platinum therapy was followed in the platinum-resistant group? (not the statistical results but only the description of the trend is ok, since on 13 used subsequent platinum therapy in the platinum-resistant group).

RESPONSE: Thank you for raising this point. To address this issue, we have performed the PFS and OS analysis according to BRCA status in the 13 patients who received subsequent platinum despite being platinum-resistant (according to our definition) after progression to PARPi. Results did not significantly differ among BRCA mutant and non BRCA-mutant patients. We have added this piece of information in the 3.3. section (lines 345-346).

  1. It is not clear who received the PARPi. It is suggested that the authors write the inclusion criteria of who received PARPi.

RESPONSE: Thank you for raising this point. We have clarified the inclusion criteria in our work in lines 171-176 and 180-181. Briefly, they are the following: (1) ovarian cancer patients of high grade serous or endometrioid histology, (2) who have received chemotherapy as per clinical practice after progression to PARPi (with at least one radiological evaluation performed), and (3) in whom PARPi had been used as maintenance after platinum therapy for relapses (regardless of BRCA mutations or other molecular tests).

In addition, it is suggested whether the HRD status were analyzed. The authors wrote “other HRR genes” and it is suggested to clarify those HRR genes. Also, please clarify whether the status of harboring the HRR genes was one of the indications of receiving PARPi and the related reference to include those patients with HRR genes.

RESPONSE: Thank you for raising this point, which was not probably clear enough in the original manuscript. In our geographical area, all patients are tested for BRCA mutations, as it is considered to a major predictive factor of response to PARPi, required for setting the therapeutic strategy. Contrary, in our geographical area, germline panels used to analyze other HRR genes were not standardized across all hospitals at the moment of the analysis and, additionally they have changed over time. Moreover, HRD status is not tested for relapses in our geographical area (currently only tested in patients undergoing front-line therapy since 2021). To sum up, knowing status of HRR genes or HRD status is no required to receive PARPi as maintenance in the relapsed setting in our geographical. Therefore, all patients included in our work have been tested for BRCA status, but not homogeneously tested of other HRR genes (nor HRD). To clarify this point, we have modified the text in page 4 line 190 (“if available”).

  1. In Supple Table2, the authors wrote “BRCA status-> others (n=4 in total)”. Please clarify who the “others” were.

RESPONSE: “other” are patients with Variants of unknown significance (VUS). We have modified Supple Table 2, and all other Tables were “others” appeared, to clarify this point.

  1. Please describe how many prior chemotherapy lines were pursued before the PARPi. In addition, please describe when each patient became platinum resistant; the patients could have become platinum resistant after 1stline of chemotherapy, or second line of platinum based-chemotherapy after the 1stplatinum based chemotherapy was sensitive. The results may differ from the ones became platinum-resistant after 1st platinum based chemotherapy and 2nd platinum based chemotherapy.

RESPONSE: We fully agree with Reviewer 2 in his/her consideration that patients who become platinum-resistant after 1st platinum-based line are a particular clinical subgroup of worse prognosis in whom rechallenge would be difficult to justify. In Table 1 we show the number of prior chemotherapy lines before PARPi, among other characteristics of the sample, in both platinum-sensitive and platinum-resistant populations. Looking into detail, all 30 platinum-resistant patients included in our work had received at least 2 prior platinum-based chemotherapy lines before PARPi (the first line and the line immediately prePARPi for relapse). Moreover, all but 3 platinum-resistant patients showed a prior TFI (months from last cycle of prior Pt to initiation of Pt immediately pre-PARPi )>6 months (also Table 1). We have clarified this in Table1.

  1. It seems the detailed information about the patients are necessary, therefore it is suggested that the raw data of patients’ clinico-genetic information be submitted.

RESPONSE: Due to restrictions for privacy and ethical requirements, data cannot be shared publicly. However, data will be available on request from the corresponding author. However, in order to describe as best as possible our population, and according to prior comments of Reviewer 2 (particularly those related to the PARPi inclusion criteria), we have further explained one feature mentioned in Table 1 but not in the first version of submitted manuscript : the sixth feature named “Months from last cycle of prior Pt to initiation of Pt immediately pre-PARPi, n (%)” aims to better characterize in which setting platinum prePARPi and PARPi had been prescribed. This feature describes the treatment-free interval between the last cycle of prior platinum to initiation of platinum immediatly pre-PARPi (which we call “prior TFI”). To this end, we have added new information in line 190-193, and in Table 1. Moreover, we are open to further clarify any other issues that may emerge, of course.

  1. Page 6 line 274-282, the authors described the figure 2. It seems that the figure 2 presents the survival curve of the patients without the 5 patients who were included in OREO study. However, page 6 line 274-281 seems to describe the ones with the 5 patients who were included in OREO study. It is suggested to clarify the sentences.

RESPONSE: Thank you for raising this point. We’ll try to clarify our message: lines 274-281 contain information from Table 3 (we have added the same abbreviations in Table 3 to clarify this). PFS and OS curves from Figure 2 are explained in lines 282-287, both text and curves excluding the 5 patients that entered OREO, as stated in Methods. As far as we understand (please correct us if needed), the last sentence of this paragraph leads to confusion “Results did not change when we excluded 5 patients with stabilization as best response to platinum-based chemotherapy prior to PARPi”. In this sentence, we aim to communicate that when we further excluded 5 patients more, the ones that had obtained stabilization after platinum-based chemotherapy prior to PARPi, results did not change. The reason why we did such complementary analysis is because we though that these 5 patients could potentially alter our results, which was not the case when we performed the analysis. To make our message clearer, we have made some modifications in lines 289-291.

  1. Page 6, line 304, the authors wrote “we identified 20 patients who could have potentially entered the OREO trial after response to subsequent platinum~~”. It means that the authors do not know who exactly were involved in the OREO trial and among them, who took “PARPi” or “placebo” as a maintenance treatment. The total of patients included in this study were 111 and 20 pts are about 18% of the total number, and it is not small. Therefore, it is suggested to identify the patients’ data related to OREO study and if it were not possible, it may be better to exclude the 20 patients.

RESPONSE: Sorry for the confusion, we should have probably been clearer in the manuscript: 20 patients met the OREO trial inclusion criteria related to the required duration of first PARPi at relapse in this trial (12 months for BRCA mutant patients or 6 months for non-mutant patients), and the duration of the subsequent platinum (at least 4 cycles). But only 5 out of 20 finally entered OREO trial (and we don’t know if they received re-olaparib or placebo), while 15 did not. We have clarified this piece of information in lines 310-314.

  1. Page 7. Line 324, the authors wrote that platinum-based regimens in platinum-resistant patients were used because the treatment-free interval was >6 months. Usually, “platinum-sensitive” is considered when the treatment-free interval was >6 months therefore, what the authors wrote in the sentence is not clear. Please, describe the definition of platinum-sensitive and platinum-resistant in this study.

RESPONSE: In this study, in order to separate patients which probably remain platinum-sensitive upon PARPi progression from those who become platinum-resistant upon progression to PARPi, we define platinum-sensitivity based on the progression-free interval (PFI) between the last cycle of the platinum-based line received immediately before the PARPi and the date of relapse to this platinum/PARPi. Although the treatment-free interval (TFI) is a usual consideration when deciding whether to retreat with platinum or not (Colombo et al, Annals of Oncology 2019), we have preferred to stick to the definition based of PFI also used in many clinical trials (Ledermann et al, Annals of Oncology 2013). We have clarified this definition in lines 203-207.

Minor:

  1. Page2, line92: preiPARPà pre PARPi

RESPONSE: Corrected. Thank you for picking up this typo error.

  1. Page6, line 277: p 0.004 àp=0.004

RESPONSE: Corrected. Thank you for picking up this typo error.

Please, don’t hesitate to contact us for further clarifications. Yours sincerely,

Round 2

Reviewer 2 Report

The authors well responded to the review.